# Protocatechuic Acid, a Simple Plant Secondary Metabolite, Induced Apoptosis by Promoting Oxidative Stress through HO-1 Downregulation and p21 Upregulation in Colon Cancer Cells

**DOI:** 10.3390/biom11101485

**Published:** 2021-10-08

**Authors:** Rosaria Acquaviva, Barbara Tomasello, Claudia Di Giacomo, Rosa Santangelo, Alfonsina La Mantia, Irina Naletova, Maria Grazia Sarpietro, Francesco Castelli, Giuseppe Antonio Malfa

**Affiliations:** 1Department of Drug and Health Science, University of Catania, Viale A. Doria 6, 95125 Catania, Italy; racquavi@unict.it (R.A.); rosrac@yahoo.it (R.S.); alfy.lamantia@gmail.com (A.L.M.); mg.sarpietro@unict.it (M.G.S.); fcastelli@unict.it (F.C.); g.malfa@unict.it (G.A.M.); 2Institute of Crystallography, National Research Council (CNR), Via Paolo Gaifami, 18, 95126 Catania, Italy; irina.naletova@ic.cnr.it

**Keywords:** CaCo-2, phenolic acids, annexin V, LDH leakage, p21, prooxidants, γ-GCS, total thiol groups

## Abstract

Gastrointestinal cancers, particularly colorectal cancer, are mainly influenced by the dietary factor. A diet rich in fruits and vegetables can help to reduce the incidence of colorectal cancer thanks to the phenolic compounds, which possess antimutagenic and anticarcinogenic properties. Polyphenols, alongside their well-known antioxidant properties, also show a pro-oxidative potential, which makes it possible to sensitize tumor cells to oxidative stress. HO-1 combined with antioxidant activity, when overexpressed in cancer cells, is involved in tumor progression, and its inhibition is considered a feasible therapeutic strategy in cancer treatment. In this study, the effects of protocatechuic acid (PCA) on the viability of colon cancer cells (CaCo-2), annexin V, LDH release, reactive oxygen species levels, total thiol content, HO-1, γ-glutamylcysteine synthetase, and p21 expression were evaluated. PCA induced, in a dose-dependent manner, a significantly reduced cell viability of CaCo-2 by oxidative/antioxidant imbalance. The phenolic acid induced modifications in levels of HO-1, non-proteic thiol groups, γ-glutamylcysteine synthetase, reactive oxygen species, and p21. PCA induced a pro-oxidant effect in cancer cells, and the in vitro pro-apoptotic effect on CaCo-2 cells is mediated by the modulation of redox balance and the inhibition of the HO-1 system that led to the activation of p21. Our results suggest that PCA may represent a useful tool in prevention and/or therapy of colon cancer.

## 1. Introduction

Epidemiological investigations indicate an increased incidence in colorectal cancer in humans worldwide [1]; in particular, nutrition plays a key role in human health and is important in determining the risk of cancer development [2,3,4,5].

It is noteworthy that gastrointestinal cancers, particularly colorectal cancer (CRC), are mostly affected by dietary factors. Several studies have revealed that almost 75% of all sporadic cases of CRC are clearly associated with poor diet and unhealthy eating habits [6,7] and that dietary modifications represent a reliable prevention strategy for reducing CRC risk [8,9]. A high intake of fresh fruits and vegetables is frequently linked to a low incidence of cancer [10]. This effect may be because these foods are rich in vitamins and phenolic compounds with antioxidant properties [11,12], such as reactive oxygen species (ROS) scavenging, electrophile scavenging, metal chelation, and inhibition of ROS generation systems. In addition, it has been reported that these compounds possess several other biological activities, including antimutagenic and/or anticarcinogenic properties [13]. Despite their antioxidant activity, phenolics show pro-oxidative potential because they can be converted into more reactive radicals or indirectly induce ROS overproduction through interaction with different molecular pathways [14]. In addition, this pro-oxidant activity allows us to sensitize cancer cells to oxidative stress by blocking their antioxidant defense, resulting in cell death through inhibition of Nrf2 pathways and, in turn, of its downstream effectors, such as antioxidant enzymes (SOD, catalase) and glutathione (GSH), thioredoxin, and HO-1 systems [15].

The heme oxygenase (HO) enzymes mainly catalyze heme degradation into the biologically active catabolites carbon monoxide (CO), biliverdin, and ferrous iron (Fe^2+^) [16].

The HO system can be triggered by various stimuli, including heme, hypoxia, lipopolysaccharides, and oxidative stress. Thus, its induction is mainly involved in inflammatory and oxidative responses. Besides its antioxidant and anti-inflammatory effects, much evidence suggests that the expression of HO-1 in cancer stimulates tumor progression through a variety of mechanisms, such as immune suppression, angiogenesis, and metastasis. However, literature data reported that HO-1 can also play an opposite antitumoral role. HO-1 is upregulated in many cancers, including colorectal, breast, melanoma, and prostate cancer. Therefore, pharmacological inhibition of HO-1 is considered a feasible therapeutic strategy in cancer treatment [17].

Many natural compounds from plants, including anthocyanins, have shown to be effective modulators of HO-1 activity via regulation of the Nrf2–HO-1 axis, whose effects on cancer cell fate depend on their concentration, cell types, and tumor microenvironment [15,18,19]. Recently, we demonstrated that an extract of *Betula etnensis* Raf. induces an oxidative cellular microenvironment resulting in CaCo-2 cell death by HO-1-mediated ferroptosis [20]. Moreover, other authors reported that downregulation of the HO-1 system is involved in mediating antiangiogenic effects of ellagic acid in prostate cancer [21] and reduces the migratory and invasive abilities of A549 cells upon resveratrol treatment [22].

Phenolic acids are secondary metabolites commonly present in plant-based food, such as fruits, vegetables, cereals, and legumes. They play a key role in several processes of plant metabolism and are strictly involved in defense mechanisms against biotic and abiotic factors [23]. Phenolic acids are mainly divided into two sub-groups, hydroxybenzoic and hydroxycinnamic, and they exhibit in vitro antioxidant activity higher than antioxidant vitamins [24]. As well as being present in numerous plant matrices, some in vitro and in vivo studies showed that protocatechuic acid (3, 4-dihydroxybenzoic acid) (PCA) is also produced as a result of anthocyanin metabolism. In fact, PCA and p-hydroxybenzoic acid were detected in rat plasma and tissues after administration of a high dose of cyanidin-3-glucoside and pelargonidin, respectively [25,26]. Among dietary hydroxybenzoic derivatives investigated as cancer preventive molecules, PCA has shown antiproliferative effects on several cancer cells in vitro, including CaCo-2 [27,28,29,30], and anthocyanins have shown antiproliferative effects on colon cancer cells in vitro [27,28,29].

As chemopreventive natural compounds that block tumor development and progression are sparking a growing clinical interest, we decided to study the potential colon-tumor-suppressive properties of PCA, demonstrating that the in vitro pro-apoptotic effect of PCA on CaCo-2 cells is mediated by the modulation of redox balance and the inhibition of the HO-1 system.

## 2. Materials and Methods

### 2.1. Chemicals

Protocatechuic acid (PCA), 3(4,5-dimethylthiazol-2-yl)2,5-diphenyl-tetrazolium bromide (MTT), and 2′,7′-dichlorofluorescein diacetate (DCFH-DA) were purchased from Sigma Aldrich (St. Louis, MO, USA). Polyclonal γ-glutamylcysteine synthetase (γ-GCS) and p21 antibodies were obtained from Abcam (Cambridge, UK). Secondary horseradish peroxidase-conjugated anti-rabbit antibody was obtained from Santa Cruz Biotechnology (Santa Cruz, CA, USA). The ELISA kit, used to measure heme oxygenase-1 (HO-1) protein concentration, was obtained from Stressgen Biotechnologies (San Diego, CA, USA). The MuseTM Annexin V & Dead Cell Kit was purchased from Merck Millipore (Billerica, MA, USA). Dulbecco’s modified essential medium and all other chemicals for cell culture were obtained from GIBCO BRL Life Technologies (Grand Island, NY, USA).

### 2.2. Cell Culture and Treatments

Human colorectal adenocarcinoma cells (CaCo-2), cultured in Dulbecco’s modified essential medium with 10% fetal calf serum, 1 mmol/L sodium pyruvate, 2 mmol/L L-glutamine, streptomycin (50 mg/mL), and penicillin (50 U/mL), were obtained from the American Type Culture Collection (Manassas, VA, USA).

The cells were plated at a constant density. After 24 h incubation at 37 °C under a humidified 5% carbon dioxide, CaCo-2 cells were treated with different concentrations of PCA (1–50–100–250–500 μM) and incubated for 72 h under the same conditions. Four replicates were performed for each sample. After the treatment, the cells were harvested by scraping, washed with PBS, and immediately utilized for the analysis.

### 2.3. MTT Bioassay

In the MTT test, used to assess cell viability, the cells were set up at 8 × 10^3^ cells per well in a 96-well flat-bottom microplate [20]. Cells were treated with PCA at different concentrations (1–50–100–250–500 μM) for 72 h. At the end of treatment time, 20 μL of 0.5% MTT in phosphate buffer saline was added to each microwell, and after 4 h of incubation with the reagent, the supernatant was removed and replaced with 100 μL of dimethyl sulfoxide. The optical density of each well sample was measured with a microplate spectrophotometer reader (Titertek Multiskan, Flow Laboratories, Helsinki, Finland) at λ = 570 nm. The percentage of viable cells was calculated relative to untreated control, considered as 100% cell viability.

### 2.4. Annexin V Determination

Muse Annexin V & Dead Cell Kit was used to evaluate apoptotic death in untreated and treated cells with PCA. Muse Annexin V & Dead Cell Reagent (100 μL) was added to the cell suspension (100 μL). Each sample, after being mixed and incubated for 20 min at room temperature in the dark, was analyzed with Muse Cell Analyzer (Millipore). The percentage of apoptotic cells was expressed as the percentage of the total cells counted for each sample.

### 2.5. Lactic Dehydrogenase Release

To assess the presence of cell necrosis as a result of cell break down, subsequent to membrane integrity leakage, we measured lactic dehydrogenase (LDH) release, as previously described [20]. Briefly, LDH activity was spectrophotometrically evaluated (U-2000 spectrophotometer; Hitachi, Tokyo, Japan) in the culture media and in the cell lysates at λ = 340 nm by monitoring NADH reduction during pyruvate–lactate transformation. Results are expressed as percentage of LDH released in the media calculated as the percentage of the total cell amount, considered as the sum of the enzymatic activity measured in the cell lysate and that in the culture media.

### 2.6. Reactive Oxygen Species Assay

ROS determination was analyzed by using a fluorescent probe 2′,7′-dichlorofluorescein diacetate (DCFH-DA), as previously described [31]. The fluorescence was evaluated spectrofluorometrically (excitation, λ = 488 nm; emission, λ = 525 nm), using an F-2000 spectrofluorometer (Hitachi). The results, compared to relative control, are reported as fluorescence intensity/mg protein. Protein content was determined using the Sinergy HT Biotech instrument by measuring the absorbance difference at λ = 280 and λ = 260.

### 2.7. Thiol Group Determination

Non-proteic total thiol groups were measured, in 200 μL of lysate supernatant, using a spectrophotometric assay based on the reaction of thiols with 2,2-dithio-bis-nitrobenzoic acid at λ = 412 nm with Sinergy HT Biotech instrument [21]. Results are expressed as nmol/mg protein, which were measured as absorbance difference at λ = 280 and λ = 260.

### 2.8. HO-1 Protein Expression

HO-1 protein expression was quantified, according to the manufacturers’ instructions, by HO-1 ELISA kit. HO-1 concentration of each sample was calculated at λ = 450 nm using a standard curve generated with purified HO-1 [32]. Results are expressed as ng/mg protein.

### 2.9. Western Blotting

The expressions of p21 and γ-glutamylcysteine synthetase (γ-GCS) were visualized by Western blot analysis as previously described [20]. The anti-p-21 (1:200 dilution) and the anti-γ-GCS (1:1000 dilution) were detected with horseradish peroxidase conjugated secondary antibody and Pierce ECL Plus substrate solution (Thermo Scientific, Rockford, IL, USA). Beta-actin was used as a loading control to normalize the expression levels of p-21 and γ-GCS proteins. The results were expressed in arbitrary densitometric units (ADU).

### 2.10. Statistical Analysis

Data were analyzed by one-way analysis of variance (ANOVA) followed by Bonferroni’s *t* test using GraphPad software 7.0 (San Diego, CA, USA). The results were obtained from three separate experiments performed in triplicate and are presented as mean ± standard deviation (SD). The value of *p* < 0.001 was considered to be significant

## 3. Results

### 3.1. PCA Cytotoxicity and Cell Death Induction

#### 3.1.1. Effects of PCA on Cell Viability of CaCo-2

Cell viability was determined by measuring the conversion of the tetrazolium salt to give color formazan, the quantity of which is proportional to the number of living cells. Figure 1 shows that the treatment of CaCo-2 with PCA (1–500 μM) for 72 h induced a significant reduction in cell viability only at high concentrations (100–500 μM), whereas, at low concentrations (1–25–50 μM), the treatment had no significant effect. Since the administration of PCA 250 and 500 μM PCA induced a similar effect, we have chosen to use 250 μM as the highest concentration for subsequent experiments.

#### 3.1.2. Annexin V Determination

To investigate the possibility that cytotoxic injury produced by PCA may be associated with apoptotic cell death, we used a cytometric analysis by cell staining with Annexin V. As reported in Figure 2, the treatment with PCA induced apoptosis in a dose-dependent fashion compared with untreated control. In particular, the percentage of apoptotic cells was markedly increased at higher doses (100 and 250 μM) by four- and five-fold, respectively. These data suggest that PCA suppresses cell viability in CaCo-2 cells through apoptotic pathways.

#### 3.1.3. LDH Release

Necrotic death, caused by disruption of the cytoplasmic membrane and the release of cytoplasmic LDH and of other cytotoxic substances into the medium, was examined by evaluating the membrane permeability of the treated cells through the existence of LDH in their culture medium. Figure 3 shows that PCA treatment (1–25–50 μM) is unable to induce LDH release, while a statistically significant increase was observed in PCA-treated CaCo-2 at 100–250 μM. These data seem to suggest that high PCA concentrations also induced necrotic cell death.

### 3.2. PCA Oxidative Properties

#### 3.2.1. Reactive Oxygen Species (ROS)

It is known that elevated ROS levels can be involved with cell death induced by various stimuli. We investigated whether PCA-induced cell death correlated with increased ROS levels. The fluorescent probe, DCFH-DA, diffused into the cells is hydrolyzed by intracellular esterases. The resulting DCFH then reacts with the intracellular oxidants to determine the observed fluorescence. The intensity of the fluorescence is proportional to the levels of the intracellular oxidizing species.

Figure 4 shows that PCA treatment ranging from 50 to 250 μM induced a significant increase in ROS levels compared to the control.

#### 3.2.2. Thiol Group Determination (RSH)

To verify the antioxidant and/or pro-oxidant capacity of PCA, we measured non-proteic total thiol group levels in CaCo-2 cells. The treatment of the cells with different concentrations of PCA (1–250 μM) induced a significant decrease in the levels of RSH; in particular, RSH levels depleted by 36% following 250 μM PCA treatment (Figure 5).

These results confirm the pro-oxidant effect exerted by the PCA in CaCo-2 cells, suggesting that the decrease in cell viability observed in the presence of PCA can be ascribed to an interference of this polyphenol not only on the antioxidant defense systems, but also on the complex network of signals involved in cell growth.

### 3.3. PCA and Proteins Expression

#### 3.3.1. γ-Glutamylcysteine Synthetase (γ-GCS) by Western Blot

The significant reduction observed in non-protein thiol groups, whose main component is glutathione, encouraged us to evaluate the expression of γ-GCS, an enzyme regulating GSH intracellular content. The Western blot analysis evidenced that the expression of γ-GCS increased in the CaCo-2 cells treated with PCA with respect to the untreated cells (Figure 6A), demonstrating that PCA promotes the onset of a pro-oxidant cell environment.

#### 3.3.2. HO-1 Expression by ELISA

Since HO-1 upregulation is involved in cancer cell proliferation and invasion, we evaluated HO-1 expression. The treatment of CaCo-2 cells with PCA resulted in a significant dose-dependent decrease in HO-1 protein expression (Figure 6B).

#### 3.3.3. p21 Expression by Western Blot

To evaluate the possible role of PCA in CaCo-2 cell proliferation and apoptosis, we evaluated its effect on p21 expression. Figure 6C shows a dose-dependent increase in p21 expression at higher concentrations (50–100–250 μM) where the expression of HO-1 was minimal.

## 4. Discussion

Several research groups have reported the anticancer activities and the sensitizing effects of various phenolics, due to their antiproliferative, antiangiogenic, and proapoptotic properties [30]. PCA is one of the major benzoic acid derivative compounds belonging to this important group of secondary metabolites, along with gallic acid, and is ubiquitously present in vegetables and fruits and as intermediates or end-products of some polyphenol metabolism [24,33]. Particularly, PCA has elicited the interest of cancer researchers because of its potential as an anti-cancer agent able to reduce proliferation, promote apoptosis, and inhibit metastasis in diverse in vitro models of cancer [34,35]. It was reported that PCA at 100 μmol/L in HepG2 and at 1–8 mM in human gastric carcinoma (AGS) cells triggered cell death and apoptosis via activation of the JNK/p38 signal [36,37]. In AGS cell line, Lin and collaborators also demonstrated that PCA exerted an antimetastatic effect by downregulation of the Ras/Akt/NF-kB pathway and consequent inhibition of MMP-2 secretion [38]. Lastly, the in vitro and in vivo anticancer effects of PCA on human breast, gastric adenocarcinoma, liver, osteosarcoma, leukemia, oral, and colon cancer cells, etc. have been observed [35,39,40,41]. The result of the present study is in agreement with these previous findings, confirming the PCA antitumor effects exerted in this study by induction of oxidative stress and apoptosis via downregulation of HO-1 and upregulation of p21. In our experimental model, the phenolic compound, starting from 100 μM, decreased CaCo-2 cell proliferation and induced apoptotic and/or necrotic cell death (Figure 1). Particularly, PCA at lower concentrations (1–25–50 μM) increased the percentage of apoptotic cells (Figure 2) without affecting cell viability. This effect might be because the Annexin V assay is based on the changes in plasma membrane lipid asymmetry with the exposure of phosphatidylserine (PS) on the outer surface of the plasma membrane bilayer, and this event represents an early biochemical apoptotic process, which does not alter mitochondrial functionality. However, a dose-dependent increase in the percentage of apoptotic cells alongside cell viability reduction and LDH leakage increment were induced at higher dosages (100 and 250 μM) (Figure 3). The necrotic effect detected by LDH assay at the highest dosages matches with previous research, demonstrating that PCA was able to induce LDH leakage through the destabilization of plasma membrane integrity [42]. To clarify the apoptotic effect shown by the PCA treatments, we analyzed the action of the compound on the oxidative state of CaCo-2 cancer cells. The involvement of ROS in apoptosis induced by different agents, such as oxidants, toxicants, or drugs, was suggested by a number of studies [43]. PCA is a powerful antioxidant agent, tenfold higher than that of the active form of vitamin E (α-tocopherol) [44]; in several cancer in vitro models, it showed both antioxidant and pro-oxidant properties [30,45,46].

Our results on the determination of ROS level (Figure 4) indicated that the cellular redox homeostasis was largely perturbed/altered towards a pro-oxidant status only by PCA 100 and 250 μM. These results suggest that in CaCo-2 tumor cell lines, PCA acts as a pro-oxidant rather than an antioxidant agent. Other studies have demonstrated that phenolic compounds including PCA with high reducing ability can not only be antioxidants but also pro-oxidants, thus generating ROS [47,48,49].

The pro-oxidant activity of PCA in CaCo-2 cells was confirmed by the steady depletion of non-protein thiol group levels at all tested concentrations (Figure 5). The free thiol residues, represented mainly by glutathione, were likely able to counteract the pro-oxidant action of PCA only at the lowest concentrations (1–25–50 μM), not at the highest concentrations where ROS levels were found to be significantly increased (100 and 250 μM). The cellular response to the condition of oxidative stress established by the treatment with PCA on CaCo-2 cells was identified by the increased protein expression levels of γ-GCS (Figure 6A), one of the major antioxidant enzymes, as it is the rate-limiting enzyme in reduced glutathione synthesis [50]. This characteristic pro-oxidant action would assume that this phenolic acid carries on business indirectly through other intracellular factors, likely targets of ROS. It was widely demonstrated that the oxidative stress and the inflammation state present in tumor cells are capable of inducing HO-1 expression through activation of various cellular signaling, including Nrf2, NF-κB, and others [51]. HO-1 in cells provides extensive tissue protection through the conversion of heme into carbon monoxide, iron, and biliverdin, which exert anti-oxidative, anti-inflammatory, and anti-apoptosis effects [52]. Moreover, HO-1 overexpression is commonly seen in several human cancers [53], including colon cancer [54], and it is required for cell cycle progression and cell proliferation in CaCo-2 cells [55]. Busserolles and collaborators previously showed that HO-1 inhibits apoptosis in colon cancer cells, thus promoting cell survival and proliferation [56]. In the present experimental model, CaCo-2 cells treated with PCA at different concentrations showed a significant decrease in HO-1 expression levels at all tested concentrations (Figure 6B). Previous studies showed that some phenolic compounds are able to inhibit the expression of HO-1 by the downregulation of the NF-kB pathway, thus promoting apoptosis [21,22,57,58]. In addition, we can speculate that HO-1 downregulation may concur to the establishment of an oxidative stress state [52] and the indirect pro-oxidant effects of PCA along with the inefficient cellular antioxidant system, which was unable to buffer the ROS overproduction. It is well known that both a shift in oxidative cell state and a downregulation of HO-1 could lead to an upregulation of the tumor-suppressor protein p53 and its downstream p21 [59], both involved in cell cycle regulation, cellular senescence, and apoptosis. Although the molecular mechanism remains unknown or controversial, a rise in p21 expression causes ROS overproduction in both normal and cancer cells [60], and it is able to mediate apoptosis in a p53-independent manner [61]. In this study, to understand the possible roles played by p21 in both ROS production and apoptosis, we evaluated the protein expression levels of p21 in PCA-treated CaCo-2 cells. Our data clearly showed that the reduced viability, ROS overproduction, and high rate of apoptosis observed at the concentrations of 100 and 250 μM were accompanied by a significant increase in p21 protein expression levels (Figure 6C), allowing us to hypothesize its involvement in the apoptotic cell death detected in this study.

Overall, our findings support that PCA at a high concentration is a pro-oxidant compound by limiting the antioxidant capacity of cancer cells. This molecule exerts its anti-tumor effects by reducing cell viability and inducing apoptosis through the modulation of a pro-oxidant cellular status by increasing ROS levels and by the inhibition of cellular antioxidant defenses and suppression of pathways leading to their expression, for instance the depletion of the intracellular pool of GSH and the downregulation of HO-1 expression. Likewise, PCA, along with the necrotic activity, can regulate the molecular pro-survival pathway, which leads to the activation of p21 and is likely mediated by HO-1 inhibition, thus promoting apoptotic death of CaCo-2 cells. In conclusion, our study supports the growing body of data that suggest the bioactivities of PCA have an impact on cancer prevention and on human health.

## Figures and Tables

**Figure 1 biomolecules-11-01485-f001:**
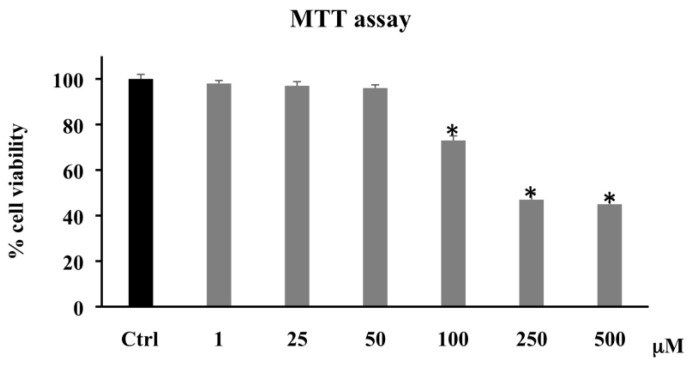
Cell viability in CaCo-2 cells untreated and treated for 72 h with PCA at different concentrations (1–500 µM). Values are the mean ± SD of four experiments in triplicate. The results are expressed as the percentage of viable cells relative to untreated control cells, considered as 100% cell viability. * Significant vs. untreated control cells: *p* < 0.001.

**Figure 2 biomolecules-11-01485-f002:**
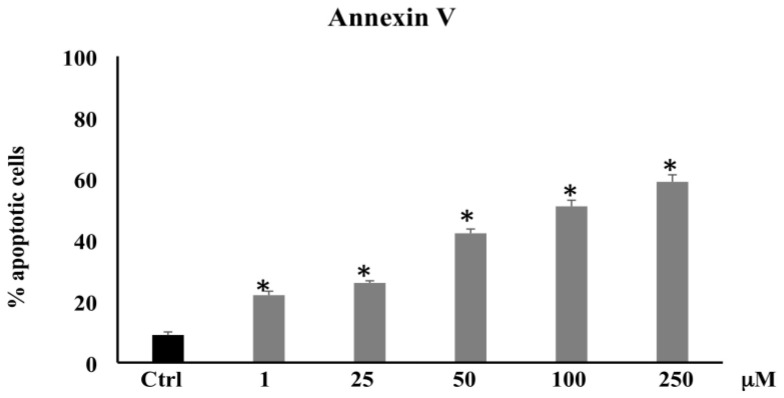
Annexin V in CaCo-2 cells untreated and treated for 72 h with PCA at different concentrations (1–250 µM). Values are the mean + SD of four experiments in triplicate. * Significant vs. untreated control cells: *p* < 0.001.

**Figure 3 biomolecules-11-01485-f003:**
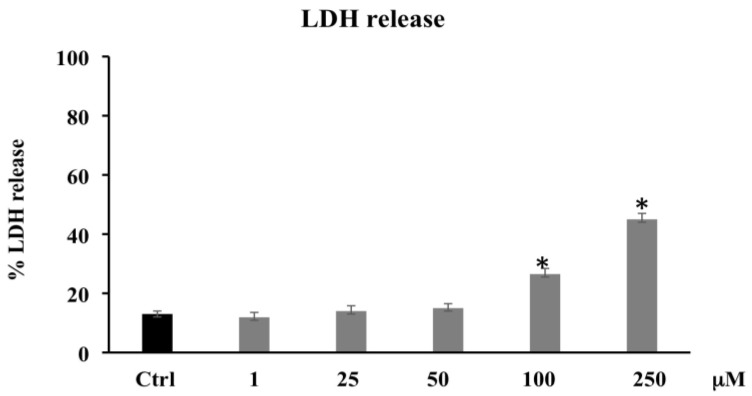
LDH released in CaCo-2 cells untreated and treated for 72 h with PCA at different concentrations (1–250 µM). Values are the mean ± SD of four experiments in triplicate. * Significant vs. untreated control cells: *p* < 0.001.

**Figure 4 biomolecules-11-01485-f004:**
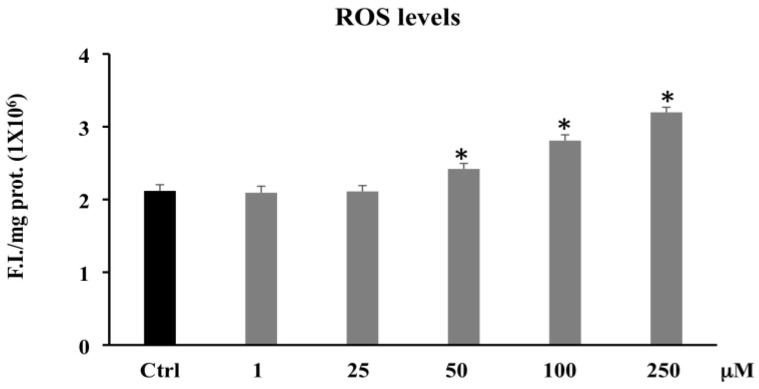
ROS levels in CaCo-2 cells untreated and treated for 72 h with PCA at different concentrations (1–250 µM). Values are the mean ± SD of four experiments in triplicate. * Significant vs. untreated control cells: *p* < 0.001.

**Figure 5 biomolecules-11-01485-f005:**
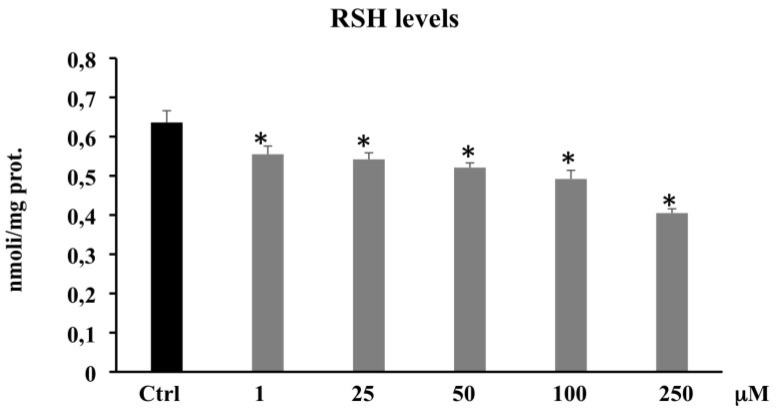
Total thiol groups in CaCo-2 cells untreated and treated for 72 h with PCA at different concentrations (1–250 µM). Values are the mean ± SD of four experiments in triplicate. * Significant vs. untreated control cells: *p* < 0.001.

**Figure 6 biomolecules-11-01485-f006:**
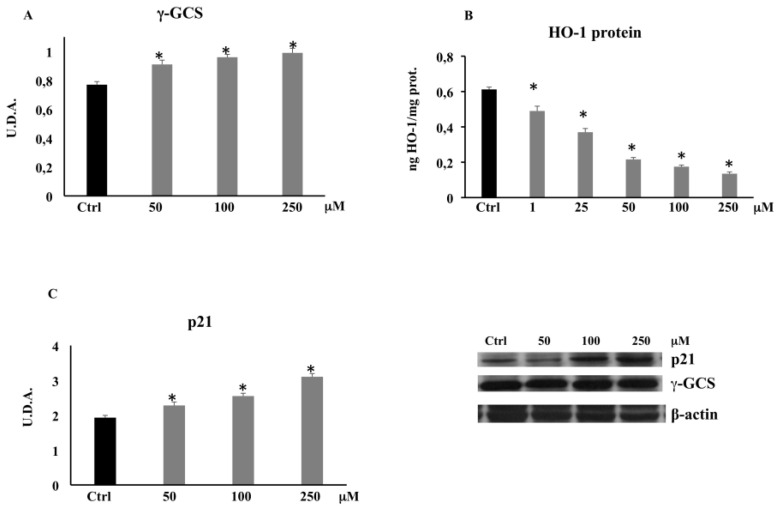
Immunoblotting of γ-GCS expression (**A**), HO-1 levels (**B**), and p21 expression (**C**) in CaCo-2 cells untreated and treated for 72 h with PCA at different concentrations (1–250 µM). Values are the mean ± SD of four experiments performed in triplicate. * Significant vs. untreated control cells: *p* < 0.001.

## Data Availability

Data were generated at Department of Drug and Health Science, University of Catania. Data supporting the results of this study are available from the corresponding authors on request.

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
