# Peer review of "Protocatechuic Acid, a Simple Plant Secondary Metabolite, Induced Apoptosis by Promoting Oxidative Stress through HO-1 Downregulation and p21 Upregulation in Colon Cancer Cells"

_biomolecules, 2021, doi:10.3390/biom11101485_

Round 1

Reviewer 1 Report

References: biomolecules-1378603

Title: „ Protocatechuic Acid, a Simple Plant Secondary Metabolite, Induced Apoptosis by Promoting Oxidative Stress through HO-1 Downregulation and p21 Upregulation in Colon Cancer Cells”

Recommendation: Minor revision

In the paper „Protocatechuic Acid, a Simple Plant Secondary Metabolite, Induced Apoptosis by Promoting Oxidative Stress through HO-1 Downregulation and p21 Upregulation in Colon Cancer Cells” by Rosaria Acquaviva et al. the authors demonstrate the effect of protocatechuic acid (PCA) on the viability, apoptosis, LDH release, ROS level, total SH content, GCS, HO-1 and p21 expression of colon cancer cells (CaCo-2 cells).

The cytotoxicity of cells proliferation by PCA was investigated by different assays depending on the concentration. Apoptosis was detected by Muse. The ROS levels were assessed by the fluorescent method. Y-GCS, HO-1, and p21 were evaluated by western blot.

  1. The experiments are planned and designed correctly. 
  2. The Materials and Methods, Results, and Discussion of this work are presented correctly.
  3. Did the Authors check the cell cultures against mycoplasma?
  4. The Authors should enlarge the font on the x and y-axis (Figure 1-6).
  5. The line 381, CRC remove.

Author Response

In the paper “Protocatechuic Acid, a Simple Plant Secondary Metabolite, Induced Apoptosis by Promoting Oxidative Stress through HO-1 Downregulation and p21 Upregulation in Colon Cancer Cells” by Rosaria Acquaviva et al. the authors demonstrate the effect of protocatechuic acid (PCA) on the viability, apoptosis, LDH release, ROS level, total SH content, GCS, HO-1 and p21 expression of colon cancer cells (CaCo-2 cells).
The cytotoxicity of cells proliferation by PCA was investigated by different assays depending on the concentration. Apoptosis was detected by Muse. The ROS levels were assessed by the fluorescent method. Y-GCS, HO-1, and p21 were evaluated by western blot. The experiments are planned and designed correctly. The Materials and Methods, Results, and Discussion of this work are presented correctly.

Authors
We would like to thank the Reviewer for such constructive comments on our manuscript. Those comments are truly helpful for revising and improving our paper. We have studied comments carefully and made corrections in the manuscript which we hope could meet with all requirements. All corrections and modifications are highlighted in red in the revised manuscript.

Question 1: Did the Authors check the cell cultures against mycoplasma?
Answer 1: Dear Reviewer thank you for your comment. We usually check the cell line cultures against mycoplasma at the beginning and at the end of the experimental protocol according to the method by the European Pharmacopeia (European Pharmacopeia. Biological Tests - Mycoplasmas. 4th ed. Strasbourg: Council of Europe Publishing; 2002. pp. 128–131.)

Question 2: The Authors should enlarge the font on the x and y-axis (Figure 1-6).
Answer 2: Thank you for pointing this out. As per your suggestion, we enlarged the font on the x and y-axis (Figure 1-6).

Question 3: The line 381, CRC remove.
Answer 3: Thank you for pointing this out. As per your suggestion, we removed CRC (line 381) in the revised manuscript. 

Reviewer 2 Report

Protocatechuic Acid, a Simple Plant Secondary Metabolite, Induced Apoptosis by Promoting Oxidative Stress through HO-1 Downregulation and p21 Upregulation in Colon Cancer Cells

This work presents the dual properties of polyphenols, as an antioxidant and pro-oxidant molecules, and their potential use in the prevention or treatment of gastric cancers, specifically in colon cancer. Particularly, the authors explore the effects of PCA treatment on CaCo-2 cells and identify the pro-oxidative and apoptotic effects of this compound, which are related to an inhibition of HO-1. Even though this work is well organized and follows a logical path, some conclusions are rash and authors should explain better how they arrive at such statements.

Abstract:

This section efficiently summarizes what has been done in this work.

Introduction:

Please, choose whether to use “antioxidant” or “antioxidative” and maintain it during the whole text.

Line 56: “OH system” please amend, and provide the definition of lps.

Line 61: “antitumor” please replace for “antitumoral”

Materials and Methods:

Line 136: “To ass the presence of cell necrosis”, please amend the typo.

Results:

Line 256: please indicate that GSH is the acronym for glutathione.

Line 259: please, when referring to Figure 6, also refer to panel C of that figure, as the western blot picture appears in that section.

Line 265: “(...) indicating that the HO-1 inhibition affects cancer growth and cell death in-vitro.” Authors jump to this conclusion but there is no causality proven between HO-1 inhibition and cancer growth and cell death. It would be optimal to include an extra experiment where HO-1 should be overexpressed.

The results presented are neat but worryingly shallow, it would be highly recommended to deepen the findings (for example with an invasion assay, by silencing or overexpressing HO-1, etc).

Overall, it would be wise to include a WB showing HO-1 expression to validate that the protein levels are decreased on account of the oxidative stress. That is not expected.

It will also be necessary to include another cell line to extend the effects showed with the present data.

Figures:

Figure 1: the legend should say that cell viability is relativized to the control.

Figure 6: please revise the legend.

Discussion:

The third paragraph of this section is difficult to read. Please check.

Line 284 reads: “Particularly, PCA at lower concentrations (1-25-50 μM) increased the percentage of apoptotic cells (Figure 2) without affecting cell viability.” So, if cell viability is not affected by the increase in apoptosis, then proliferation might be responsible for the absence of differences in this measurement. Authors must discuss these effects in the discussion section.

The last sentence reads: “Concluding, our study supports the growing body of data suggesting the bioactivities of PCA and its potential impact on cancer therapy and on human health.”. Before mentioning the PCA potential as cancer therapy, it could be of interest to address certain questions. Are the cytotoxic effects of PCA on non-tumoral cells known? Has this compound already been used clinically?

Author Response

Protocatechuic Acid, a Simple Plant Secondary Metabolite, Induced Apoptosis by Promoting Oxidative Stress through HO-1 Downregulation and p21 Upregulation in Colon Cancer Cells

This work presents the dual properties of polyphenols, as an antioxidant and pro-oxidant molecules, and their potential use in the prevention or treatment of gastric cancers, specifically in colon cancer. Particularly, the authors explore the effects of PCA treatment on CaCo-2 cells and identify the pro-oxidative and apoptotic effects of this compound, which are related to an inhibition of HO-1. Even though this work is well organized and follows a logical path, some conclusions are rash and authors should explain better how they arrive at such statements. Abstract: This section efficiently summarizes what has been done in this work.

AUTHORS 

We would like to thank the Reviewer for such constructive comments on our manuscript. Those comments are truly helpful for revising and improving our paper. We have studied comments carefully and made corrections in the manuscript which we hope could meet with all requirements. All corrections and modifications are highlighted in red in the revised manuscript.

Introduction:

Question 1: Please, choose whether to use “antioxidant” or “antioxidative” and maintain it during the whole text.

Answer 1: Thank you for pointing this out. We harmonized the words according to your suggestion throughout the whole text (page 1 line 43)

Question 2: Line 56: “OH system” please amend, and provide the definition of lps.

Answer 2: Thank you for pointing this out. We corrected the typo and provided the full name for lps. (page 1, line 56).

Question 3: Line 61: “antitumor” please replace for “antitumoral”.

Answer 3: Thank you for pointing this out. We replaced the word according to your suggestion (page 2, line 61)

Materials and Methods: 

Question 4: Line 136: “To ass the presence of cell necrosis”, please amend the typo.

Answer 4: Thank you again for pointing this out. We corrected the typo. (page 3, line 117).

Results:

Question 5: Line 256: please indicate that GSH is the acronym for glutathione.

Answer 5: Thank you for your comment, we indicated that GSH is the acronym for glutathione in the text line 52 (page 2) and in the Abbreviations list (page 9, line 388)

Question 6: Line 259: please, when referring to Figure 6, also refer to panel C of that figure, as the western blot picture appears in that section.

Answer 6: Thank you for pointing this out, we modified text and figure according to your suggestion. (page 7, lines 264 and 268; page 8 line 325)

Question 7: Line 265: “(...) indicating that the HO-1 inhibition affects cancer growth and cell death in-vitro.” Authors jump to this conclusion but there is no causality proven between HO-1 inhibition and cancer growth and cell death. It would be optimal to include an extra experiment where HO-1 should be over expressed.

Answer 7: Dear Reviewer thank you for your comment. We agree with you that this sentence in the results section is surely a stretch. We eliminated what you highlighted. However, we found your suggestion very interesting, and a second part of the study is underway in which HO-1 overexpression, inhibition or silencing are under consideration as well as further biological tests such as invasion assay and also the addition of another cell line as you suggested later (question 8 and 10).

Question 8: The results presented are neat but worryingly shallow, it would be highly recommended to deepen the findings (for example with an invasion assay, by silencing or overexpressing HO-1, etc).

Answer 8: Dear Reviewer thank you for your comment. As reported in the previous point (answer 7) we found your suggestions very interesting and we’ll take them into account in the second part of the study which is underway.

Question 9: Overall, it would be wise to include a WB showing HO-1 expression to validate that the protein levels are decreased on account of the oxidative stress. That is not expected.

Answer 9: Dear Reviewer thank you for your comment. The human HO-1 elisa kit we used in our experimental model was designed for the quantitative measurement of Heme Oxygenase 1 protein in cell culture extracts, cell culture supernatant, serum, tissue extracts etc. It is fully validated in biological samples and guaranteed by the manufacturer to possess high sensitivity, specificity and reproducibility comparable to western blot.

Question 10: It will also be necessary to include another cell line to extend the effects showed with the present data.

Answer 10: Dear Reviewer thank you for your comment, please see answer 7 and 8

Figures:

Question 11: Figure 1: the legend should say that cell viability is relativized to the control.

Answer 11: Thank you for pointing this out, the legend of figure 1 was revised

Question 12: Figure 6: please revise the legend.

Answer 12: Thank you for pointing this out, the legend of figure 6 was revised

Discussion:

Question 13: The third paragraph of this section is difficult to read. Please check.

Answer 13: Thank you for pointing this out, the third paragraph in the discussion section was revised according to your suggestion (page 8, lines 297-301)

Question 14: Line 284 reads: “Particularly, PCA at lower concentrations (1-25-50 μM) increased the percentage of apoptotic cells (Figure 2) without affecting cell viability.” So, if cell viability is not affected by the increase in apoptosis, then proliferation might be responsible for the absence of differences in this measurement. Authors must discuss these effects in the discussion section.

Answer 14: Annexin V assay is based on the changes in plasma membrane lipid asymmetry with the exposure of phosphatidylserine (PS) on the outer surface of the plasma membrane bilayer. This event represents an early biochemical apoptotic process, which does not alter mitochondrial functionality; the MTT test (which we used for cell viability) measures mitochondrial functionality, that is not modified by the low concentrations of PCA which, on the other hand, are able to initiate the apoptotic process. In fact, it is known that PS externalization occurs before the loss of mitochondrial integrity and probably this event is closely correlated with the absence of cell death at the lowest concentrations as detected in our experimental model.( PMID: 20032867, PMID: 22858544, PMID: 24418990, PMID: 31665027)

Question 15: The last sentence reads: “Concluding, our study supports the growing body of data suggesting the bioactivities of PCA and its potential impact on cancer therapy and on human health.”. Before mentioning the PCA potential as cancer therapy, it could be of interest to address certain questions. Are the cytotoxic effects of PCA on non-tumoral cells known? Has this compound already been used clinically?

Answer 15: Dear Reviewer thank you for your comment. We based our conclusion not only on the reported data but also on the considerable scientific literature available in the field (PMID: 33389946, PMID: 16407799, PMID: 32872307, PMID: 25356681, PMID: 20840540, PMID: 32659939, PMID: 19601677). However, we agree with you about the lack of more in-depth studies including clinical studies to date, therefore we revised the last sentence of the conclusions.

Round 2

Reviewer 2 Report

All the concerns have been addressed. The present form of the manuscript is improved